# Design, Synthesis, Characterization, and Evaluation of the Anti-HT-29 Colorectal Cell Line Activity of Novel 8-Oxyquinolinate-Platinum(II)-Loaded Nanostructured Lipid Carriers Targeted with Riboflavin

**DOI:** 10.3390/pharmaceutics15031021

**Published:** 2023-03-22

**Authors:** Tugce Boztepe, Sebastián Scioli-Montoto, Rocio C. Gambaro, María Esperanza Ruiz, Silvia Cabrera, José Alemán, Germán A. Islan, Guillermo R. Castro, Ignacio E. León

**Affiliations:** 1Laboratorio de Nanobiomateriales, CINDEFI—Departamento de Química, Facultad de Ciencias Exactas, Universidad Nacional de La Plata-CONICET, La Plata B1900, Argentina; 2Consejo Nacional de Investigaciones Científicas y Técnicas (CONICET), La Plata B1904, Argentina; 3Laboratorio de Investigación y Desarrollo de Bioactivos (LIDeB), Departamento de Ciencias Biológicas, Facultad de Ciencias Exactas, Universidad Nacional de La Plata (UNLP), La Plata B1900, Argentina; 4Instituto de Genética Veterinaria (IGEVET, UNLP-CONICET La Plata), Facultad de Ciencias Veterinarias Universidad Nacional de La Plata (UNLP), La Plata B1900, Argentina; 5Departamento de Química Inorgánica, Universidad Autónoma de Madrid, 28049 Madrid, Spain; 6Departamento de Química Orgánica, Universidad Autónoma de Madrid, 28049 Madrid, Spain; 7Max Planck Laboratory for Structural Biology, Chemistry and Molecular Biophysics of Rosario (MPLbioR, UNR-MPIbpC), Partner Laboratory of the Max Planck Institute for Biophysical Chemistry (MPIbpC, MPG), Centro de Estudios Interdisciplinarios (CEI), Universidad Nacional de Rosario, Rosario S2000, Argentina; 8Nanomedicine Research Unit (Nanomed), Center for Natural and Human Sciences (CCNH), Universidade Federal do ABC (UFABC), Santo André 09210-580, SP, Brazil; 9CEQUINOR (UNLP, CCT-CONICET La Plata, Asociado a CIC), Departamento de Química, Facultad de Ciencias Exactas, Universidad Nacional de La Plata, La Plata B1900, Argentina; 10Cátedra de Fisiopatología, Departamento de Ciencias Biológicas, Facultad de Ciencias Exactas, Universidad Nacional de La Plata, La Plata B1900, Argentina

**Keywords:** drug delivery, colorectal cancer, platinum, riboflavin

## Abstract

Colorectal cancer is occasionally called colon or rectal cancer, depending on where cancer begins to form, and is the second leading cause of cancer death among both men and women. The platinum-based [PtCl(8-O-quinolinate)(dmso)] (8-QO-Pt) compound has demonstrated encouraging anticancer activity. Three different systems of 8-QO-Pt-encapsulated nanostructured lipid carriers (NLCs) with riboflavin (RFV) were investigated. NLCs of myristyl myristate were synthesized by ultrasonication in the presence of RFV. RFV-decorated nanoparticles displayed a spherical shape and a narrow size dispersion in the range of 144–175 nm mean particle diameter. The 8-QO-Pt-loaded formulations of NLC/RFV with more than 70% encapsulation efficiency showed sustained in vitro release for 24 h. Cytotoxicity, cell uptake, and apoptosis were evaluated in the HT-29 human colorectal adenocarcinoma cell line. The results revealed that 8-QO-Pt-loaded formulations of NLC/RFV showed higher cytotoxicity than the free 8-QO-Pt compound at 5.0 µM. All three systems exhibited different levels of cellular internalization. Moreover, the hemotoxicity assay showed the safety profile of the formulations (less than 3.7%). Taken together, RFV-targeted NLC systems for drug delivery have been investigated for the first time in our study and the results are promising for the future of chemotherapy in colon cancer treatment.

## 1. Introduction

Colorectal cancer (CRC) is the third most common type of cancer globally, after lung and breast cancers (1.93 million new cases in 2020), and the second most deadly [1]. While overweight or obesity, certain types of diets, physical inactivity, and alcohol and tobacco use are shown as risk factors, CRC is inherited in only 5% of the cases [2].

Metallodrugs are a class of anticancer agents mostly used in the treatment of several pathologies including diabetes, neurodegenerative diseases, and cancer [3,4,5,6]. The most successful metallodrugs are cisplatin, carboplatin, and oxaliplatin, widely used in the treatment of several tumors and various kinds of cancer including colorectal, ovarian, cervical, testicular, lung, head, and neck cancers [7,8]. Despite their effective clinical applications, long-term use can cause drug resistance, dose-limiting side effects, and undesirable side effects in non-target tissues and organs [9]. In previous work, 8-oxyquinolinate-platinum(II) [PtCl(8-O-quinolinate)(dmso)] (8-QO-Pt) exhibited higher antitumor activity compared to cisplatin, generating neither resistance nor side effects in an osteosarcoma model in mice [10].

To overcome the drawbacks of platinum drugs, different drug delivery systems have been developed [9,11]. Nanosystems can enhance the bioavailability of these drugs by increasing the residence time in the bloodstream and protecting them from degradation while preventing possible damage to healthy tissue and delivering the drug in a controlled manner. Therefore, potential side effects and resistance to anticancer drugs may be averted [12]. Nanostructured lipid carriers (NLCs) are second-generation solid lipid nanoparticles (SLNs) that provide improved drug loading capacity and stability [13]. They are synthesized by the mixing of a solid lipid phase with a liquid lipid (at room temperature), and the further addition of a surfactant that acts as a dispersant and stabilizing agent [14].

Drug delivery systems can be roughly classified as passive or active. Passive drug delivery is allowed by the enhanced permeability and retention effect (EPR) that is characterized by the higher blood capillary permeability in the tumor tissue with a much lesser return of the fluids to the lymphatic circulation [15]. However, passive drug delivery based on the EPR effect has limitations. A dense fibrotic microenvironment can cause inhibition of the deep internalization of nanosystems [16]. On the other hand, active targeting delivery is based on a system that releases the drug into a targeted area [17]. The integration of active targeting ligands into the nanosystems has been sought to increase nanoparticle accretion at the tumor site [18]. Antibodies, antibody fragments, peptides, proteins, aptamers, and receptor ligands can be utilized to create an active drug delivery system [15]. In particular, numerous targeted nanosystems for delivering platinum-based drugs have been reported [9].

Riboflavin (RFV) or vitamin B2 is a water-soluble molecule (XlogP3 = −1.5) available in many foods and used as a common dietary supplement. RFV participates in the energetic and respiratory metabolism of cells through the synthesis of the two major flavin coenzymes (i.e., flavin adenine dinucleotide and flavin mononucleotide) [19]. Previous research has shown that RFV transporters and the riboflavin carrier protein show up-regulation in various tumor types such as breast, prostate, and hepatocellular carcinoma [20,21], as well as in some colon carcinoma cells [19]. Considering these results, RFV could be an efficient targeting ligand for tumor-specific drug delivery. To date, several different nanodelivery systems have been functionalized with RFV, such as liposomes [20], polymer conjugates [22], and telodendrimers [23], to actively target tumor cells.

The present work aimed to design, synthesize, characterize, and evaluate in vitro antitumoral activity of a new RFV-functionalized 8-QO-Pt-loaded NLC for the treatment of colon cancer. Three formulations, which differ in the phase in which RFV was incorporated during the nanoparticle synthesis processes, were prepared. The nanosystem characterization included dynamic light scattering (DLS) and transmission electron microscopy (TEM) analysis. In addition, drug encapsulation efficiency, drug release and modeling, cell viability, cellular uptake, apoptosis, and hemotoxicity analyses of the nanoparticles were carried out. To the best of our knowledge, this is the first report of RFV-targeted NLCs for the delivery of a platinum compound.

## 2. Materials and Methods

### 2.1. Materials

Myristyl myristate (MM) and capric triglyceride lipids were kindly donated by Croda (Martinez, Argentina). Poloxamer 188, riboflavin, and 3,3′-dioctadecyloxacarbocyanine perchlorate (DiOC18) were purchased from Sigma-Aldrich (Buenos Aires, Argentina). Dulbecco’s modified Eagle’s medium (DMEM) and TrypLE™ were purchased from Gibco (Gaithersburg, MD, USA). Fetal bovine serum (FBS) was bought from Internegocios S.A. (Mercedes, Argentina). Annexin V, Fluorescein isothiocyanate (FITC), propidium iodide (PI), and tetrazolium salt MTT (3-(4,5-dimethylthiazol-2-yl)-2,5-diphenyl-tetrazolium-bromide) were supplied by Invitrogen Co. (Buenos Aires, Argentina). Other reagents were of analytical or HPLC grade from available commercial sources and used as received from Merck (Darmstadt, Germany) or a similar brand.

### 2.2. Methods

#### 2.2.1. Preparation of the 8-QO-Pt

The 8-QO-Pt was synthesized and characterized according to Martín Santos et al. [24], obtaining a yellow-orange solid (78% yield).

^1^H NMR (300 MHz, CD_2_Cl_2_) δ: 9.40 (dd, *J* = 10.7, 1.1 Hz, 1H), 8.37 (dd, *J* = 8.3, 1.0 Hz, 1H), 7.57–7.40 (m, 2H), 7.05–7.02 (m, 2H), 3.62 (s, 6H).

#### 2.2.2. Preparation of the Formulations

The 8-QO-Pt-compound-loaded (+) formulations of NLC/RFV were synthesized by the ultrasonication technique [25]. Different ratios of RFV were added to the lipidic or the aqueous phases according to Table 1. For the first formulation (R_1_-8-QO-Pt-NLC), 400 mg of myristyl myristate (MM) (2.0%, *w*/*v*) was melted in a water bath at 70 °C and mixed with 2.0 mg of the 8-QO-Pt compound, previously dissolved in 200 μL capric triglyceride (1.0%, *v*/*v*). Then, 20 mL of a hot aqueous solution with 10 mg of RFV (0.05%, *w*/*v*) and Poloxamer 188 (4.5%, *w*/*v*) was added to the lipid phase. Immediately, the mixture was ultrasonicated at 65% amplitude for 30 min using a 6 mm titanium tip probe (ultrasonic processor, Cole Parmer, USA, 130 W). After sonication, the formulation was cooled down at room temperature and stored at 4 °C for further studies. On the other hand, in the second formulation (R_2_-8-QO-Pt-NLC), the RVF (10 mg) was incorporated directly into the lipid phase, whereas in the third one (R_3_-8-QO-Pt-NLC), 5 mg of RVF was incorporated in the aqueous phase and 5 mg of RFV into the lipid phase before mixing.

#### 2.2.3. Measurement of Encapsulation Efficiency

The encapsulation efficiency (EE, %) of 8-QO-Pt in the nanosystems was determined by the indirect method according to Equation (1). Briefly, a volume of 500 µL of each formulation was transferred to a cutoff centrifugal filter (MWCO 10,000, Microcon^®^, Millipore, MA, USA) and centrifuged at 12,000× *g* for 30 min. The 8-QO-Pt in the filtrate was measured by HPLC (see Section 2.2.6). The EEs of the formulations were calculated as follows:EE (%) = (w_i_ − w_fd_)/w_i_ × 100 (1)
where w_i_ is the initial amount of 8-QO-Pt compound added to the formulation, and w_fd_ is the amount of non-encapsulated 8-QO-Pt compound in the filtrate after the ultrafiltration process.

#### 2.2.4. Particle Size, Polydispersity Index, Zeta Potential, and Transmission Electron Microscopy

The average diameter and particle size distribution of NLCs were measured by dynamic light scattering (Nano ZS Zetasizer, Malvern Instruments Corp, UK) at 25 °C in polystyrene cuvettes with a thickness of 10 mm. Measurements were carried out in 10 mm path length capillary cells, using deionized water (Milli-Q^®^, Millipore, MA, USA). The polydispersity Index (PdI) values and the zeta potentials (ζ) were also determined. All the measurements were performed in triplicate to obtain the mean value.

TEM analysis was conducted using a Jeol-1200 EX II-TEM microscope (Jeol, MA, USA). First, the NLC formulations were diluted 10 times with ultrapure Milli-Q^®^ water, and 10 μL of the sample was spread onto a collodion-coated Cu grid (400-mesh). Liquid excess was taken out with filter paper, and for contrast enhancement, a drop of phosphotungstic acid was added to the samples.

#### 2.2.5. In Vitro Drug Release Assay

In vitro drug release assay was performed using Float-A-Lyzer^®^G2 dialysis devices (MWCO: 100 kD). The dialysis devices were previously soaked in 10% ethanol for 10 min and then left in distilled water for 20 min. Next, the devices were filled with 2.0 mL of each formulation and immersed in 15 mL of 30% isopropyl alcohol solution at 37 °C, with continuous shaking at 200 rpm [26,27]. For the free drug release, 2.5 mg of 8-QO-Pt was dissolved in 25 mL of the release medium and 2.0 mL of the solution was transferred to the dialysis device. Then, 200 μL samples were withdrawn at regular intervals for 24 h, and drug concentration was measured by HPLC.

#### 2.2.6. HPLC Analysis

Chromatographic analysis was performed using HPLC (Gilson SAS, Villiers-Le-Bel, France) with UV-VIS detection and a Zorbax Eclipse XDB-C18 (150 mm × 4.6 mm, 5 μm, Agilent Technologies Inc., Santa Clara, CA, USA) column as the stationary phase. The mobile phase consisted of a mixture of methanol and water (55:45). The system was operated isocratically at a 0.6 mL/min flow rate and the detection was performed at 262 nm. Samples were diluted with mobile phase and centrifuged (15,000× *g* for 5 min at 4 °C) before their injection (20 μL). The linearity, precision, and specificity of the method were validated over the range of expected concentrations (0.25–50.00 μg/mL).

#### 2.2.7. Cell Cytotoxicity Assay

The human colon carcinoma cell line (HT-29) was purchased from ATCC (HTB-38™) and cultured in DMEM (Gibco, Invitrogen Corporation, USA) supplemented with 10% FBS (Internegocios, Buenos Aires, Argentina) and antibiotics (100 U/mL penicillin and 100 μg/mL streptomycin; Gibco, Invitrogen Corporation, USA) at 37 °C and under a 5% CO_2_ atmosphere. Cell cytotoxicity assay was performed using the 3-(4,5-dimethylthiazol-2-yl)-2,5-diphenyltetrazolium bromide (MTT) reagent [28]. Cells were seeded in a 96-multiwell plate and grown at 37 °C for 24 h. For the cytotoxicity assay, cells were treated with different concentrations (2.5, 5.0, 7.5, and 10.0 µM) of free 8-QO-Pt, R_1_-8-QO-Pt-NLC, R_2_-8-QO-Pt-NLC, and R_3_-8-QO-Pt-NLC in serum-free DMEM for 24 h. After this treatment, the cells were washed with PBS and incubated with 0.5 mg/mL MTT in supplemented DMEM for 3 h. Next, the absorbance was read spectrophotometrically in a microplate reader (multiplate reader multiscan FC, Thermo Scientific, MA, USA) at 570 nm. Cell viability was expressed as the percentage (%) of the untreated control value (100% survival).

#### 2.2.8. Cellular Uptake Assay

The cellular uptake assay was performed using empty (−) formulations of NLC/RFV (R_1_-NLC, R_2_-NLC, and R_3_-NLC) loaded with the green fluorescent probe DiOC18 (λabs/λem = 484/501 nm) by adding 1.0 mg of DiOC18 to the lipid phase at 70 °C. The DiOC18-labeled formulations were synthesized by ultrasonication as mentioned above. The DiOC18 was 100% incorporated into the NLCs. Flow cytometry was utilized to evaluate the internalization of the labeled nanoparticles into the HT-29 cells. Cells were seeded on a 12-well plate to be allowed to attach to the bottom surfaces of the wells for 24 h. Later, the cells were treated with 2.5 µM of the formulations for 24 h. After 24 h of treatment, the cells were washed with PBS and treated with 300 μL of trypLE until they were unattached from the surface. Next, 600 μL of the serum-containing medium was placed in each well and the cells were transferred to Eppendorf tubes and centrifuged at 2500× *g* at 4 °C for 5 min. Supernatants were separated carefully and the pellets were washed with PBS once. Later, the pellets were dispersed in 200 μL of PBS. Fluorescence was analyzed by FACSCalibur (Becton Dickinson, Franklin Lakes, NJ, USA), and the values were read by FlowJo 7.6 software.

#### 2.2.9. Apoptosis Assay

Early and late apoptotic cells were determined by annexin V-FITC and PI staining. The cells were seeded on a 12-well plate for 24 h and treated with two different concentrations (2.5 and 5.0 µM) of the formulations for 24 h. After the treatment and for the staining, the cells were washed with PBS and diluted with 1X binding buffer, Annexin V-FITC, and PI, and incubated at room temperature for 20 min before the analysis. Cells were collected using flow cytometry (BD FACSCalibur™) and the results were analyzed using FlowJo 7.6 software. For each analysis, 10,000 counts gated on an FSC vs. SSC dot plot were recorded. Four subpopulations were defined in the dot plot: the undamaged vital (FITC−/PI−), the vital mechanically damaged (FITC−/PI+), the early apoptotic (FITC+/PI−), and the late apoptotic (FITC+/PI+) subpopulations.

#### 2.2.10. Hemotoxicity Assay

Venous blood was obtained from healthy donors after written informed consent and collected in heparinized tubes. Then, whole blood was diluted in a six-well plate with culture medium Ham F12 with 10% FBS in a final volume of 2.0 mL. Each culture was subjected to different treatments with free 8-QO-Pt, free RFV, their combinations (8-QO-Pt and RFV), and 8-QO-Pt-loaded or -unloaded formulations of NLC/RFV according to each specific experimental design, and kept in the culture at 37 °C with 5% CO_2_ for 24 h and 48 h. The released hemoglobin was quantified by the absorbance read at 540 nm in a TECAN-infinite M200 Pro spectrophotometer to determine the percentage of lysed erythrocytes. Total hemolysis (100%) was achieved by incorporating Triton X-100 into the medium, while the physiological solution was used as a negative control.

#### 2.2.11. Statistical Analysis

All experiments were carried out with a minimum of three independent replicates. Comparisons of the means were performed by analysis of variance (ANOVA) with a significance level of 5.0% (α = 0.05) followed by Fisher’s least significant difference test.

## 3. Results and Discussion

### 3.1. Formulation Development and Nanoparticle Morphology

All the formulations were prepared following a procedure previously reported by our group [25,29]. The lipid matrices were composed of myristyl myristate (MM, solid lipid), capric triglyceride (liquid lipid), the surfactant Poloxamer 188, and the RFV phase, and the content was changed according to the formulation. The formulations under study were: R_1_-8-QO-Pt-NLC containing 10 mg of RFV in the aqueous phase; R_2_-8-QO-Pt-NLC, containing 10 mg of RFV in the lipid phase; and finally, R_3_-8-QO-Pt-NLC containing 5 mg of RFV in the aqueous phase and 5 mg of RFV in the lipid phase. The preparations were made before the mixing and ultrasonication. All three 8-QO-Pt-loaded formulations showed an EE(%) higher than 70%, that is, 74.6 ± 1.7, 79.4 ± 1.1, and 79.5 ± 2.3 for R_1_-8-QO-Pt-NLC, R_2_-8-QO-Pt-NLC, and R_3_-8-QO-Pt-NLC, respectively.

The mean diameter, PdI, and Z-pot (ζ) values of the formulations (with and without the drug) are listed in Table 2. All the formulations showed a narrow size distribution with average particle sizes from 144.50 ± 2.04 to 175.00 ± 1.18 nm. RFV did not significantly affect the particle size of 8-QO-Pt-loaded (+) formulations of NLC/RFV. The particle sizes were substantially monodisperse, with PdI values around 0.2, except for R_2_-NLC, which was higher than the optimal maximal limit of 0.3 [30,31]. The ζ value is a measure of the effective electric charge on the surface of the nanoparticles and is also one of the indicators of the stability of the colloidal systems [32]. The 8-QO-Pt-loaded (+) formulations exhibited lower negative ζ than the controls. The ζ values tend to decrease in the presence of a platinum compound in the formulations [33].

TEM was used to evaluate the morphology and size distribution of the nanoparticles. TEM images showed that the nanoparticles had a spherical shape. Moreover, the incorporation of 8-QO-Pt and RFV did not influence the morphology of nanoparticles (Figure 1).

### 3.2. In Vitro Drug Release Assay

In vitro release studies of 8-QO-Pt from the NLC were carried out by the dialysis method using 30% isopropyl alcohol solution as the release medium. An amount of 2 mL of each formulation was transferred into the Float-A-Lyzer^®^ G2 dialysis device and the release of 8-QO-Pt from the nanoparticles was analyzed as a function of time. The in vitro release experiments could predict the stability of the formulations in terms of drug delivery once they reached physiological environments. Several studies have investigated the relationship between formulation stability and release profile in order to select the best nanoparticulate systems to be potentially administered [34,35,36]. According to the results shown in Figure 2, it was observed that the R_2_-8-QO-Pt-NLC and R_3_-8-QO-Pt-NLC formulations showed instability as soon as they were exposed to the release medium since NLC started releasing 8-QO-Pt as fast as the free drug. On the other side, the R_1_-8-QO-Pt-NLC exhibited a more controlled release profile. However, it is important to mention that the in vivo release profile of NLC will be modulated by interactions with different molecules from the bloodstream, which may form a “protein corona” that could delay the release of the cargo molecules [37]. All the formulations were able to effectively release the drug in the assayed conditions. In particular, R_1_-8-QO-Pt-NLC reached 62% of drug release after 24 h, while R_2_-8-QO-Pt-NLC and R_3_-8-QO-Pt-NLC showed a release profile similar to the free drug, reaching over 80% of the content released at that time. Since R_1_-8-QO-Pt-NLC was the only formulation with a significantly different release than the control, the DDSolver Excel add-in program [38] was used to fit different kinetic models that could provide some insights about the release behavior. The analysis showed that for the R_1_-8-QO-Pt-NLC formulation, the Baker and Lonsdale model best fitted the data, with an R^2^_adj_ of 0.9364. This model, derived from Higuchi’s model, is generally used to describe the controlled release of drugs from spherical matrices, and diffusion and degradation are the main factors responsible for the release mechanism [39]. This type of device comprises a dispersion of the solid drug through the rate-controlling medium (i.e., the matrix). For low levels of drug loading, as in this case, the release involves the dissolution of the drug in the matrix followed by diffusion to the surface of the device. Therefore, the release profile is determined by the initial loading, the chemical nature of the matrix, and its geometry [40].

### 3.3. Cell Cytotoxicity Assay

The in vitro cytotoxic activity against the HT-29 colon carcinoma cell line was investigated by MTT assay. Figure 3 presents the cell viability (%) with 2.5, 5.0, 7.5, and 10.0 µM concentrations of the free 8-QO-Pt compound and 8-QO-Pt-loaded (+) formulations of NLC/RFV after 24 h treatment. The results revealed that both the free and loaded 8-QO-Pt (in all three formulations), displayed a dose-dependent cytotoxicity profile against HT-29 cells while the formulations without 8-QO-Pt did not present any toxicity. The cell viability (%) was reduced to 70.7 ± 4.7% (free 8-QO-Pt), 59.7 ± 2.3% (R_1_-8-QO-Pt-NLC), 78.3 ± 5.6% (R_2_-8-QO-Pt-NLC), and 91.2 ± 4.8% (R_3_-8-QO-Pt-NLC) for 2.5 µM; 67.5 ± 6.3%, 17.7 ± 2.8%, 31.8 ± 4.4%, and 51.4 ± 4.7% for 5.0 µM; 29.0 ± 4.6%, 8.7 ± 0.1%, 11.1 ± 0.8%, and 16.1 ± 2.7% for 7.5 µM; and 12.9 ± 1.0%, 8.6 ± 0.3%, 10.8 ± 4.5%, and 9.8 ± 2.4% for 10 µM, respectively. Even though the three formulations contained the same amount of RFV (10 mg in total), the incorporation of RFV into distinct phases affected cell viability, particularly at low doses (2.5 and 5.0 µM). All the RFV-targeted 8-QO-Pt-loaded formulations demonstrated a higher cytotoxic effect than the free 8-QO-Pt compound (except at 2.5 µM). Moreover, R_1_-8-QO-Pt-NLC seemed to be more effective, especially at 5.0 µM, showing higher antitumor activity compared to free 8-QO-Pt, R_2_-8-QO-Pt-NLC, and R_3_-8-QO-Pt-NLC (3.8-fold, 1.8-fold, and 2.9-fold, respectively). Moreover, Table 3 lists the IC_50_ values for the free 8-QO-Pt compound and the nanoformulations in the HT-29 cells. In our previous study, the 8-QO-Pt-NLC nanosystem without RFV functionalization did not show a superior cytotoxic effect in comparison to the free 8-QO-Pt in HT-29 cells [29]. These results reveal that the RFV functionalization of nanoparticles increases cytotoxicity by targeting delivery [23]. As a third-generation of platinum-based anticancer drug, oxaliplatin has been approved to treat CRC [9]. The IC_50_ value of oxaliplatin in the HT-29 cells was determined as 2 µM [4] and another study has shown a value of 5 µM after 48 h treatment [41]. It has been reported that the IC_50_ value of oxaliplatin-loaded solid lipid nanoparticles was found to be higher than in folic-acid-functionalized oxaliplatin-loaded SLN but was lower than in free oxaliplatin in the HT-29 cells [42]. Our findings demonstrate that the R_1_-8-QO-Pt-NLC presents a similar effective antiproliferative activity profile in the HT-29 colorectal cancer cell line as the reference drug currently used in the clinic (oxaliplatin).

### 3.4. Cellular Uptake Assay

To assess the cellular internalization of the NLC/RFV nanoparticles, the fluorescent probe DiOC18 was incorporated into all empty (−) formulations. Then, HT-29 cells were treated with empty labeled nanoparticles at a concentration equivalent to 2.5 µM of 8-QO-Pt-loaded (+) formulations of NLC/RFV for 24 h (Figure 4). After the treatment, the fluorescent signals were detected by flow cytometry. According to the results, HT-29 cells were able to successfully internalize the nanoparticles. The mean ± standard deviation of cell uptake demonstrated that R_1_-8-QO-Pt-NLC showed the highest cellular uptake (89.5 ± 1.0%) in comparison to R_2_-8-QO-Pt-NLC (25.6 ± 1.1%) and R_3_-8-QO-Pt-NLC (47.7 ± 1.5%). In addition, the untreated condition (basal media) showed 2.0 ± 0.4%. These data could be correlated with the superior cytotoxic effect of R_1_-8-QO-Pt-NLC observed in Figure 3.

### 3.5. Apoptosis Assay

While anticancer drugs cause cell apoptosis, some biochemical and morphological changes can be observed with the use of tracers. The cell membrane exposes phosphatidylserine residues on the outer surface that can be detected with the fluorescent dye Annexin V-FITC by fluorescence assays [43]. Therefore, HT-29 cells were stained with the Annexin V-FITC/PI apoptosis detection kit after 24 h of treatment with free 8-QO-Pt and R_1_-8-QO-Pt-NLC, at 2.5 µM and 5.0 µM. Figure 5 shows the percentages of vital (FITC−/PI−), early apoptotic (FITC+/PI−), late apoptotic (FITC+/PI+), and necrotic (FITC−/PI+) subpopulations in the dot plot. According to the results, the free 8-QO-Pt compound and R_1_-8-QO-Pt-NLC induced 16.4 ± 3.7% and 20.8 ± 1.2% (at 2.5 µM), and 22.5 ± 4.8% and 37.9 ± 9.9% (at 5 µM) of cells in late apoptosis (FITC+/PI+), respectively. The fact that R_1_-8-QO-Pt-NLC was able to generate more apoptosis can be explained by taking into consideration that this formulation was the one with the highest cell internalization ability, i.e., it can deliver the 8-QO-Pt drug more efficiently into the cells. In our previous study, untargeted 8-QO-Pt-NLC and free 8-QO-Pt showed very similar results in the apoptosis assay [29]. When the same formulation was functionalized with RFV, the antiproliferation effect of 8-QO-Pt increased due to the higher affinity of RFV as a targeting agent. It has been shown that the conjugation of RFV with some nanosystems such as ultrasmall iron oxide nanoparticles, polyethylene glycol polymers, dendrimers, and liposomes exhibited high affinity toward tumors in preclinical studies [44]. However, the previous RFV-conjugated systems displayed some disadvantages. Iron oxide nanoparticles are well-known promoters of free radicals that could damage the surrounding tissues [45]. PEG polymers were reported to produce retarded immune responses in several patients [46]. Dendrimers showed high tissue toxicity beyond G3 (i.e., third generation) [47], and liposome stability requires constant size and size distribution for a prolonged time which may cause the liposomes to be unable to effectively release their cargo into the targeted cells [48].

### 3.6. Hemotoxicity Assay

The hemocompatibility of drug delivery nanosystems becomes the first approach to determine potential cytotoxic effects after the interaction of the blood cells with nanoparticles [49]. Considering a possible systemic administration of platinum nanoparticles by the intravenous route, the effect of the nanoparticles and their components on whole blood was studied (Figure 6). The hemolysis process could be related to the nature of the particles, the presence of surfactants, or strong negative charges on the surface [50,51]. In the case of 8-QO-Pt, a dose-dependent hemolytic effect was observed but produced a hemolysis degree of less than 1.5% at the highest tested concentration (5.0 µM) at 24 h and 48 h (Figure 6a). Higher hemotoxicity values for RFV were found, with hemolysis percentages around 2.0%, 3.0%, and 4.5% for the 7.5, 15.0, and 30.0 µM concentrations, respectively. Combinations of 8-QO-Pt and RFV at different proportions showed that the hemotoxic effect of RVF overlapped with the cytotoxic effect of 8-QO-Pt with no additive and/or synergetic effects being observed. On the other hand, the hemolysis degree of 8-QO-Pt-loaded (+) and empty (−) formulations of NLC/RFV after exposure to human blood cells for 24 h and 48 h was determined (Figure 6b). The hemotoxicity decreased for the formulations containing 8-QO-Pt and RFV compared to combinations of free 8-QO-Pt and RFV. A dose–hemolytic effect response in the case of 8-QO-Pt-loaded NLCs and a reduced hemolysis percentage in the case of empty NLCs (less than 1.0% in all cases) were observed. The R_1_-8-QO-Pt-NLC formulation showed hemolysis around 3.0%, R_2_-8-QO-Pt-NLC around 2.0%, and R_3_-8-QO-Pt-NLC around 3.7% at the concentration of 5.0 µM. These results suggest that R_2_-8-QO-Pt-NLC is the safest and R_3_-8-QO-Pt-NLC the most hemotoxic formulation. Nevertheless, all the formulations are in an acceptable and low range of hemotoxicity (lower than 5%), which is the value considered as toxic according to the ISO/TR 7406 [52].

## 4. Conclusions

In summary, RFV ligand-targeted NLC particles have been revealed as potential tumor-specific drug delivery systems. The current study reports an efficient encapsulation strategy of 8-QO-Pt into three novel formulations of NLC/RFV, which differ in the way RFV was incorporated into them. DLS and TEM analysis showed that the nanoparticles had a spherical shape and small particle size with a narrow size distribution (144–175 nm). A sustained release of 8-QO-Pt from the NLC/RFV nanoparticles was observed which could reduce the possible adverse effects caused by systemic administration of the drug. The formulations exhibited a dose-dependent manner in the cytotoxicity assays. In particular, the antitumor effect of the R_1_-8-QO-Pt-NLC system was superior to that of the free 8-QO-Pt compound and the other tested nanosystems. On the other hand, the cellular internalization level of R_1_-8-QO-Pt-NLC was found to be higher than those of R_2_-8-QO-Pt-NLC and R_3_-8-QO-Pt-NLC. In comparison to the free drug, the apoptosis assay revealed that active targeting with RFV resulted in a great antiproliferative effect due to cancer cell selectivity. Moreover, the high hemocompatibility of the nanoparticles was proven by the hemotoxicity assay which is an advantage in the case of intravenous administration. This relevant result makes the R_1_-8-QO-Pt-NLC system a good candidate for further in vivo studies.

Finally, RFV-targeted 8-QO-Pt-NLC systems for drug-controlled release have been developed and investigated in our study for the first time. The results are superior to previous RFV-targeted anticancer systems. The RFV-targeted 8-QO-Pt-NLC can be a potentially promising nanosystem for the chemotherapy of colorectal cancers.

## Figures and Tables

**Figure 1 pharmaceutics-15-01021-f001:**
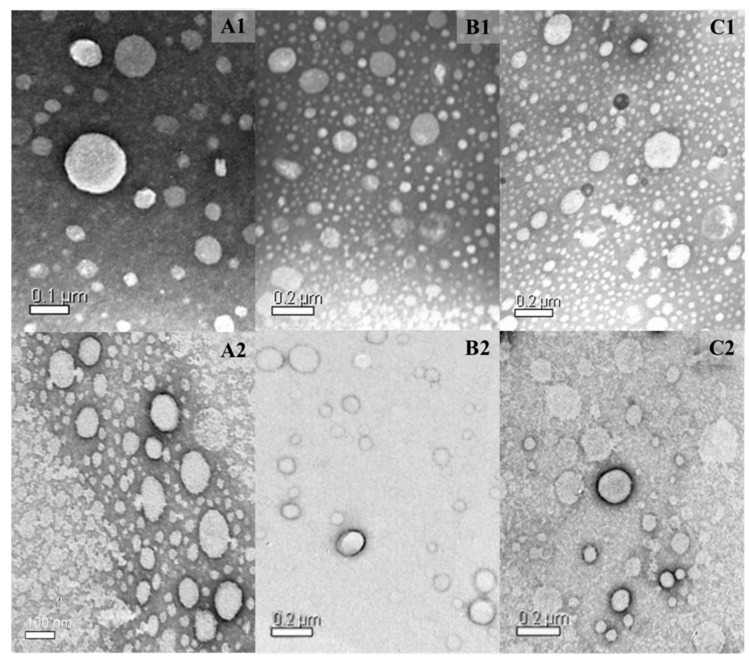
TEM images of the nanoparticles of NLC/RFV: R_1_-8-QO-Pt-NLC (**A1**), R_1_-NLC (**A2**), R_2_-8-QO-Pt-NLC (**B1**), R_2_-NLC (**B2**), R_3_-8-QO-Pt-NLC (**C1**), and R_3_-NLC (**C2**).

**Figure 2 pharmaceutics-15-01021-f002:**
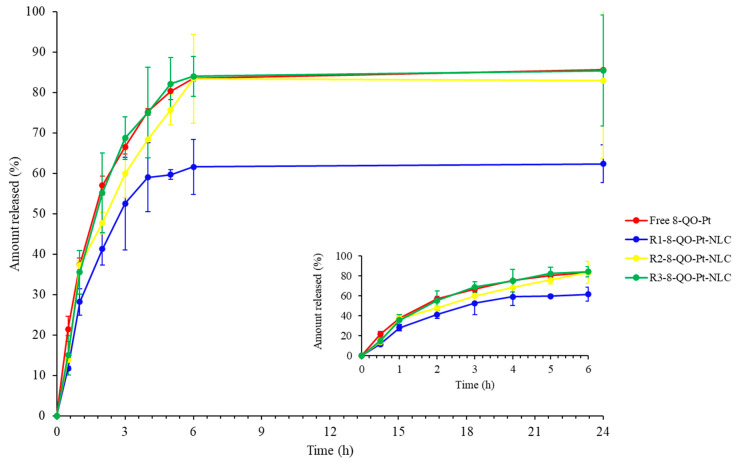
Release profiles (mean ± SD) of free 8-QO-Pt (red), R_1_-8-QO-Pt-NLC (blue), R_2_-8-QO-Pt-NLC (yellow), and R_3_-8-QO-Pt-NLC (green) for 24 h.

**Figure 3 pharmaceutics-15-01021-f003:**
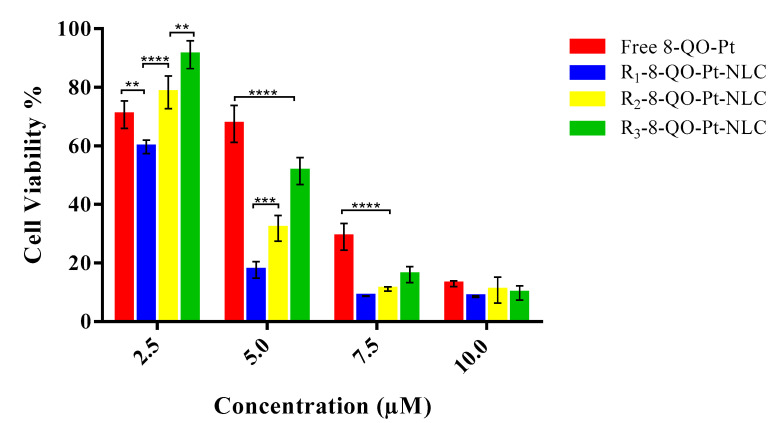
Cell cytotoxicity of the free 8-QO-Pt compound and 8-QO-Pt-loaded (+) formulations of NLC/RFV against the HT-29 cell line for 24 h. Data are shown as mean ± SD (*n* = 3). (** *p* < 0.001; *** *p* < 0.0003; **** *p* < 0.0001).

**Figure 4 pharmaceutics-15-01021-f004:**
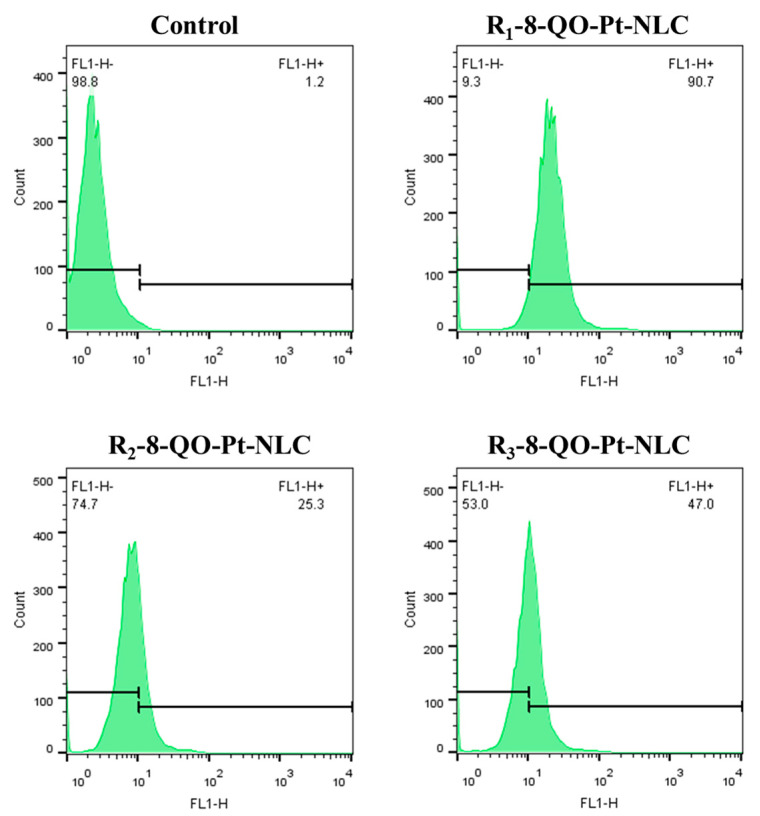
Cellular uptake of DiOC18-loaded empty (−) nanoparticles of NLC/RFV in HT-29 cells measured by flow cytometry. The cells were incubated for 24 h at a concentration equivalent to 2.5 μM of 8-QO-Pt-loaded (+) formulations of NLC/RFV.

**Figure 5 pharmaceutics-15-01021-f005:**
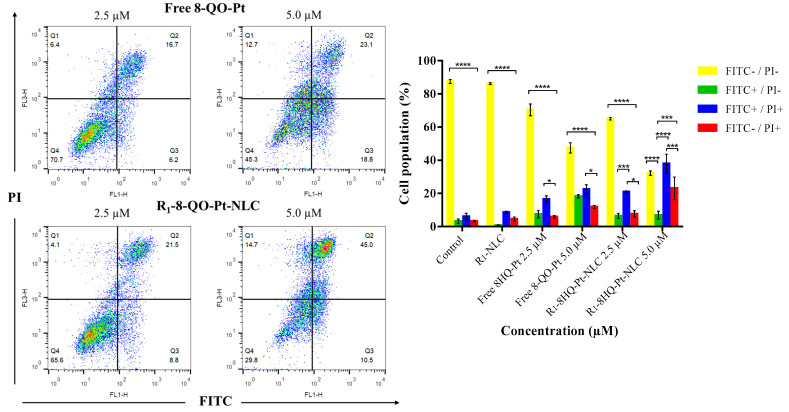
Apoptosis effect of the free 8-QO-Pt compound and R1-8-QO-Pt-NLC at 2.5 μM and 5.0 μM after 24 h incubation with HT-29 cells. Data are shown as mean ± SD (*n* = 3). (* *p* < 0.01; *** *p* < 0.0002; **** *p* < 0.0001).

**Figure 6 pharmaceutics-15-01021-f006:**
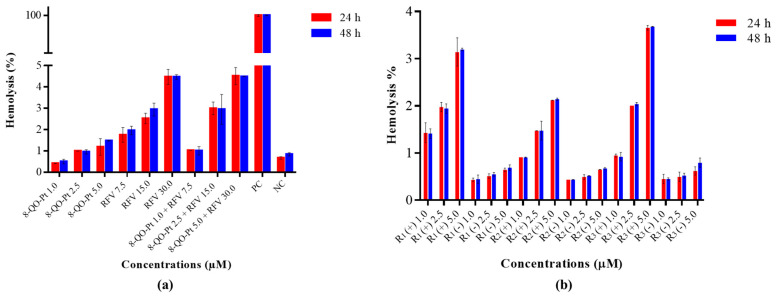
(**a**) Hemolysis degree of 8-QO-Pt, RFV, and their combinations after exposure to human blood cells for 24 h and 48 h. Controls: positive (PC) with 1.0% Triton X-100 and negative (NC) with untreated blood. (**b**) Hemolysis degree of 8-QO-Pt-loaded (+) and empty (−) formulations of NLC/RFV after exposure to human blood cells for 24 h and 48 h. Data are shown as mean ± SD.

**Table 1 pharmaceutics-15-01021-t001:** The amount of RFV in the formulations.

Formulations	RFV Phase (mg)	
	Aqueous	Lipid
R_1_-8-QO-Pt-NLC	10.0	-
R_2_-8-QO-Pt-NLC	-	10.0
R_3_-8-QO-Pt-NLC	5.0	5.0

**Table 2 pharmaceutics-15-01021-t002:** Zav, PdI, and Z-pot of NLCs obtained by DLS. Zav: average size (nm); PdI: polydispersity index; Z-pot: ζ (mV). All values are the mean and standard deviation (*n* = 3).

Formulations	Zav (nm)	PdI	ζ (mV)
R_1_-NLC	162.6 ± 6.6	0.285 ± 0.030	−6.14 ± 0.53
R_2_-NLC	175.0 ± 1.2	0.354 ± 0.040	−9.39 ± 1.69
R_3_-NLC	145.7 ± 1.4	0.236 ± 0.010	−13.30 ± 0.70
R_1_-8-QO-Pt-NLC	146.5 ± 1.6	0.227 ± 0.000	−5.75 ± 0.51
R_2_-8-QO-Pt-NLC	149.3 ± 0.8	0.243 ± 0.000	−8.27 ± 0.19
R_3_-8-QO-Pt-NLC	144.5 ± 2.0	0.233 ± 0.010	−11.70 ± 0.20

**Table 3 pharmaceutics-15-01021-t003:** IC_50_ values (µM) of free 8-QO-Pt and the formulations after 24 h of incubation.

	IC_50_ Value (µM)
Free 8-QO-Pt	5.3
R_1_-8-QO-Pt-NLC	2.9
R_2_-8-QO-Pt-NLC	3.9
R_3_-8-QO-Pt-NLC	5.0

## Data Availability

Not applicable.

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
