# Peer review of "Design, Synthesis, Characterization, and Evaluation of the Anti-HT-29 Colorectal Cell Line Activity of Novel 8-Oxyquinolinate-Platinum(II)-Loaded Nanostructured Lipid Carriers Targeted with Riboflavin"

_pharmaceutics, 2023, doi:10.3390/pharmaceutics15031021_

Round 1

Reviewer 1 Report (Previous Reviewer 1)

The authors have not addressed all issues satisfactorily. I insist that this manuscript is unsuitable to be published in Pharmaceutics.

Author Response

In our last coverletter, we answered point by point the inquiries of Reviewer 1. Unfortunately, we have not had an answer by reviewer 1.

Reviewer 2 Report (Previous Reviewer 2)

Most of my comments were addressed, however there is still room for improvement.

1. The title of the paper may be misleading as it suggests that tested  compounds were active against tumor or in vivo. Since, biological part of the study was conducted using only one cell line HT-29 in vitro I recommend to rephrase the title and replace "the anti-tumoral activity" with "anti HT-29 colorectal cell line cells activity".

2. Inconsistencies in data presenting, in Fig 1, 3 and 5 results are presented as mean ± SEM, while in Fig. 4 as mean ± SD (See text lines 352-353: "mean ± standar desviation of cell uptake demonstrated that R1-8-QO-Pt-NLC...").

In the legend of Fig. 2 such information is not given at all.

Author Response

Thanks for the fruitful comments of reviewer 2.

According to the reviewer´s comments, we have been changed the title in and we have been normalized the data presented in the figures in the new version of the manuscript.

This manuscript is a resubmission of an earlier submission. The following is a list of the peer review reports and author responses from that submission.

Round 1

Reviewer 1 Report

This manuscript describes 8-QO-Pt encapsulated nanostructured lipid carriers (NLC) with riboflavin (RFV) for treating colorectal cancer. However, the experimental methods of this study are not rigorous enough, and the data are not presented well. Overall, this manuscript is unsuitable to be published in Pharmaceutics, as the following issues should be considered.

1. As oxaliplatin, the first platinum-based compound approved for colorectal cancer, oxaliplatin solution should be hired as positive control for in vitro and in vivo antitumor activity evaluation.

2. The in vivo studies, which are necessary to elucidate the antitumor effects of RFV-functionalized 8-QO-Pt-loaded NLCs, were not included in this study.

3. When it comes to Figure 1, there appears to be some crystals in the TEM images, indicating that the 8-QO-Pt encapsulated NLCs may not stable.

4. In Figure 3, it would be better to draw dose–response curves instead of bar graphs, and it is suggested to compare the IC50 of each formulation with positive control.

5. According to Figure 2, the release of RFV-functionalized 8-QO-Pt-loaded NLCs is too fast, so the authors need to provide more evidence that how long 8-QO-Pt encapsulated NLCs could remain stable in the blood circulation after administration.

Reviewer 2 Report

This is a well-written paper by Boztepe and co-workers evaluating anti-tumoral activity of novel 8-Oxyquinolinate-platinum(II)-loaded Nanostructured Lipid Carriers targeted with Riboflavin.

Authors documented the morphology and size distribution of the formulated nanoparticles by DLS and TEM; evaluated in vitro drug release profiles; assessed cytotoxic activity against colorectal cancer cell line HT-29; evaluated cellular internalization of riboflavin-targeted nanoparticles’ formulation; measured the potency of  8-Oxyquinolinate-platinum(II)-loaded nanostructured lipid carriers targeted with riboflavin to trigger apoptosis (Annexin V binding) and hemocompatibility of these drug delivery formulations.

The authors concluded that nanoparticles have a spherical shape and small particle size with narrow size distribution (144-175 nm) and such 8-QO-Pt encapsulation allow sustained release drug from the NLC/RFV nanoparticles. Particularly, the antitumor effect of the R1-8-QO-Pt-NLC system was superior to the free 8-QO-Pt compound and the other tested nanosystems. Out of the free formulations tested, the cellular internalization level of R1-8-QO-Pt-NLC was found to be highest. In comparison to the free drug, the apoptosis assay displayed that active targeting with RFV resulted in an elevated antiproliferative effect due to higher cancer cell selectivity. Furthermore, 8-QO-Pt-NLCs were shown to be safe and hemocompatible as proven by the hemotoxicity assay, thus made the R1-8-QO-Pt-NLC system a good candidate for further in vivo studies. Finally, the authors suggested that the RFV guided 8-QO-Pt-NLC is superior to non-targeted 8-QO-Pt-NLC and thus can be a potential promising nanosystem for chemotherapy of colon cancers.

This is an interesting study, methodologically and clinically valuable.The study was well performed and the data were thoroughly analyzed and interpreted. The authors extensively discussed the results. The idea and results presented in this manuscript may attract attention of some groups of researchers, especially those searching for new anti-cancer modalities.

There are a few minor issues that need to be addressed:

Comment 1. Experiment of NLC formulations cytotoxicity against HT-29 included only 4 samples tested: free 8-QO-Pt compound and 8-QO-Pt loaded (+) formulations of NLC/RFV. In my opinion this toxicity experiment requires also testing of nanoparticles without drug 8-QO-Pt, namely NLC and NLC + riboflavin to see non-drug related toxicity or side effects of nanoparticles.

Comment 2. In my opinion it is hard to assess cytotoxicity of the tested drug objectively by using only one cell line. Although, this is rather basic pharmacological study, such limited use of cells with different characteristics seems not justified.

Comment 3. Please indicate in the Figures legend or in 2.2.11. Statistical Analysis section whether you show mean values ± SD or SEM on the graphs.

Comment 4. If the cellular uptake were performed more than once the results of the experiments (Fig. 4) could be quantified, average of independent experiments calculated and presented on the graph. It would be beneficial for readers to compare values among formulations (statistical significance).

Taken together, this paper by Boztepe and co-workers represent a worthwhile contribution to cancer research. It could be accepted with modifications as suggested. I recommend the manuscript for further publication process.